# The *Azadirachta indica* (Neem) Seed Oil Reduced Chronic Redox-Homeostasis Imbalance in a Mice Experimental Model on Ochratoxine A-Induced Hepatotoxicity

**DOI:** 10.3390/antiox11091678

**Published:** 2022-08-28

**Authors:** Galina Nikolova, Julian Ananiev, Veselin Ivanov, Kamelia Petkova-Parlapanska, Ekaterina Georgieva, Yanka Karamalakova

**Affiliations:** 1Department of Chemistry and Biochemistry, Medical Faculty, Trakia University, 11 Armeiska Str., 6000 Stara Zagora, Bulgaria; 2Department of General and Clinical Pathology, Forensic Medicine and Deontology, Faculty of Medicine, Trakia University, 11 Armeiska Str., 6000 Stara Zagora, Bulgaria

**Keywords:** *A. indica* oil, OTA, GSPx, Hyd, GST, PGC-1, STIR-1, cytokines

## Abstract

Liver damage severity depends on both the dose and the exposure duration. Oxidative stress may increase the Ochratoxine-A (OTA) hepatotoxicity and many antioxidants may counteract toxic liver function. The present study aims to investigate the hepatoprotective potential of *Azadirachta indica*
*A* (*A. indica*; neem oil) seed oil to reduce acute oxidative disorders and residual OTA toxicity in a 28-day experimental model. The activity of antioxidant and hepatic enzymes, cytokines and the levels of oxidative stress biomarkers –MDA, GSPx, Hydroxiproline, GST, PCC, AGEs, PGC-1, and STIR-1 were analyzed by ELISA. The free radicals ROS and RNS levels were measured by EPR. The protective effects were studied in BALB/C mice treated with *A. indica* seed oil (170 mg/kg), alone and in combination with OTA (1.25 mg/kg), by gavage daily for 28 days. At the end of the experiment, mice treated with OTA showed changes in liver and antioxidant enzymes, and oxidative stress parameters in the liver and blood. *A. indica* oil significantly reduced oxidative stress and lipid peroxidation compared to the OTA group. In addition, the hepatic histological evaluation showed significant adipose tissue accumulation in OTA-treated tissues, while treatment with 170 mg/kg *A. indica* oil showed moderate adipose tissue accumulation.

## 1. Introduction

The liver, a major detoxifying organ in the human body, is involved in the biosynthesis and metabolism of macromolecules such as proteins, lipids, and carbohydrates [1]. Redox imbalance in macromolecular biosynthesis leads to liver diseases which can be inflammatory, non-inflammatory, and degenerative, all with the possibility of liver fibrosis progression [1]. Chronic liver damage caused by mycotoxins induces redox-homeostatic up-regulation in liver cells and interrupts liver function and hepatocyte stimulation that activates apoptotic signaling [2]. The synthesis of different mycotoxins genera is carried out by sequential oxidative, alkylating, or condensing enzymatic reactions in the metabolism of malonate, mevalonate, acetate, or amino acids [3,4]. The most toxic fungal agent produced by Aspergillus ochraceus and Penicillium verrucosum fungi is Ochratoxin A (OTA) [5,6]. Once in the organs, OTA disrupts the antioxidant-prooxidant balance, causing oxidative stress (OS) disorders, which in turn activate key mechanisms in gene expression, and genetically determine the physiological cell death processes [7]. Due to accumulated cellular mycotoxicosis, OTA is associated with hepatotoxic, carcinogenic, and other effects affecting biological systems and cells [8,9]. According to Pfohl-Leszkowicz et al. [10], OTA affects blood coagulation and carbohydrate metabolism. The research of Petkova-Bocharova et al. [11] provided evidence for the involvement of OTA in Balkan endemic nephropathy, and the development of urinary tract tumors. Imaoka et al. [12] reported acute hepatotoxicity induced by OTA and concluded that OTA metabolism varies among different tissues. An in vivo study by Gagliano et al. [13] highlighted the fact that the overall early hepatic OTA effects are different from the nephro effects, as the toxin does not play a fibrogenic role. OTA-induced acute liver injury contributes to the weakening of the body’s ability to counteract the redox-homeostatic imbalance and, in particular, to stop altered gap junctional intercellular communication (GJIC). In vitro studies discussed the fact that GJIC is directly affected by cellular OS disorders, and changes in the expression of GJIC and E-cadherin contribute to the progression of tumorigenesis [13]. In addition, chronic exposure to OTA leads to hepatocarcinoma development [12,13].

*Azadirachta indica A.* (*A. indica*; neem) leaves, bark, fruits, flowers, resin, and oil contain valuable components (alkaloids, flavonoids, phenols, etc.) that are used in the treatment of hypertension, heart disease, diabetes, and cancer [14]. *A. indica* oil has a high percentage of antioxidant activity due to its total phenolic content and great free radical scavenger ability [15]. The combined use of *A. indica* seed oil with mycotoxins may reduce redox homeostatic imbalances and related side effects and possibly provide symptomatic relief. Moreover, mycotoxin accumulated OS in biological systems can be neutralized by the active ingredients contained in *A. indica* seed oil [16]. The increased hepatic redox-homeostatic imbalance produced by reactive oxygen species (ROS) overproduction leads to a decrease in the endogenous and exogenous antioxidant protection of the hepatocyte cell, which has a multistage system.

This cellular “vulnerability” is expressed in the presence of low molecular weight intra- and extracellular antioxidants, and the limited activity of antioxidant enzymes. [17]. Oxidative disorders induce damage to the structure of nucleic acids, lipids, and proteins, and impair the synthesis of prostaglandins, leukotrienes, and thromboxanes [18]. In addition, the generation of secondary active forms stimulates radical development [19]. Free radical production is part of normal cellular function but increased redox-homeostatic imbalance damages all types of macromolecules [20], and produces hepatotoxicity [21], and liver fibrosis [22].

The main hypothesis is that *A. indica* seed oil regulates the OS redox-homeostatic imbalance and the immunomodulatory response after OTA-induced acute liver inflammation in experimental mouse models, in part, through its protective effects on liver tissue and full restoration of liver function. Sirtuin 1 (SIRT1) is a member of the proteins from Sirtuins family. SIRT1 functions together with peroxisome proliferator-activated receptor gamma coactivator 1-α (PGC-1α) and encode the nicotinamide adenine dinucleotide (NAD+)-dependent histone deacetylase. SIRT1 interacts with PGC-1α and deacetylates it, resulting in increased activity, which induces hepatic gene transcription. Moreover, SIRT1 plays a central role in metabolic function, and increased ROS levels as a consequence of mitochondrial function [23]. According to Lagouge et al., SIRT1 regulates liver metabolism by peroxisome proliferator-activated receptor-gamma (PPARγ)-coactivator 1 alpha (PGC-1α) [24]. SIRT1 protects against OS by activating PGC-1α gene transcription and regulates the expression of antioxidant enzymes (SOD and glutathione peroxidase) and the inflammatory response. In addition, SIRT1 can control the transcription of genes, such as interleukin (IL)-1, tumor necrosis factor α (TNF-α), IL-10, IL-6, and other inflammatory factors by regulating the level of acetylation [25].

The specific objectives were to investigate the protective effect of *A. indica* seed oil on biomarkers of oxidative stress through the oxidation products of proteins and lipids, the activity of antioxidant enzymes, and the production ROS and RNS in the liver and blood. A correlation analysis was performed between the OS biomarkers and the studied biochemical parameters. We hypothesize that *A. indica* oil prevented the OTA-induced liver disorders by regulating the protein carbonyl content (PC), protein peroxidation, and protein expression of the SIRT1-mediated PGC-1α pathway, inhibiting the production of cytokines and decreasing oxidative disorders.

## 2. Materials and Methods

### 2.1. Plant Material

*A. indica* (seeds) was collected from the Eastern province (Mechi Anchal) of Nepal in the end of 2021. The *A. indica* seeds were decorticated manually to obtain kernels, which were then spread over filter paper and left to air dry for about 96 h to a constant weight. The dry kernels were ground in a coffee mill and portions of the material were used for further investigation. The oil was extracted by an 8 h Soxhlet extraction with petroleum ether. *A. indica* oil was extracted from the *Azadirachta indica A* seed according to the methodology of Deng et al. [26]. All chemicals were analytical grade (AR > 99.7%). The used OTA was donated by prof. St Stoev, Trakia University, Stara Zagora, Bulgaria [27].

### 2.2. Animals, Experimental Design, and Ethical Approval

BALB/C male mice (*n* = 24) aged 6 weeks (mean weight 27–36 ± 2.0 g) were purchased from the Institute of Animal Science, Slivnitsa, Bulgaria. After transport, the animals had a 10-day cycle of adaptive feeding and acclimatization. During the experiment, they had ad libitum free access to fresh water and food, maintained 12 h/d light/dark cycles, 23 ± 2 °C and humidity 55% in the vivarium of the Faculty of Medicine of Trakia University. The work procedures were approved by the Trakia University Animal Ethics Commission, Stara Zagora, Bulgaria, and the Bulgarian Food Safety Agency, Sofia, Bulgaria with a license (325/6000-0333/09.12.2021) following the Directive 2010/63/EU on the animals’ protection used for experimental and other scientific work.

Figure 1 shows the protocol of the experiment where the experimental animals were divided into four groups (*n* = 6) under controlled environmental conditions: (1) control group (*n* = 6) on a basic diet (containing 19.6% crude protein, 4.03% crude fat, 6.89% crude fiber, 10.71% moisture and 8.97% ash); (2) *A. indica* oil group (*n* = 6) administered by gavage for 28 consecutive days (170 mg/kg b.w.); (3) OTA group (*n* = 6) administered by gavage (1.25 mg/kg b.w.) per day given by gavage via a feeding needle for 28 consecutive days; (4) *A. indica* seed oil (*n* = 6) (170 mg/kg b.w.) and OTA (1.25 mg/kg b.w.) once daily for 28 days. The used *A. indica* seed oil and OTA were mixed with d.H_2_O and crude olive oil (Lekkas Farm, Mikro Horio, Greece) before treating the mice. The animal feed was tested, and no other mycotoxins were found. The physiological state and behavior of the experimental animals were monitored daily.

### 2.3. Dissection Procedure

Mice were individually daily weighed for a period of 28 days (presented as mean ± SEM for each group) and were measured for food intake and body weight (B.W.). Their general physical condition and food intake were monitored. The animals were sacrificed under anesthesia (Nembutal 50 mg/kg i.p.) on day 29 from the beginning of the experiment. Blood samples were collected by intracardiac standard technique and fresh blood was collected with vacutainer serum tubes. Serum samples were prepared by centrifugation (4000 rpm, 10 min, 4 °C). Liver samples were stored in ice-cold PBS (4 °C), weighed (SW), and homogenized.

### 2.4. Histological Analysis for Visualization of Liver Changes

The preparation of the liver samples for histopathological examination included removal and perfusing of the right liver, followed by immersing the tissues in a fixative 10% aqueous formalin solution for 24 h. After dehydration in a graduated series of ethanolic concentrations, the blocks are clarified in xylene and embedded in paraffin. The tissue 4 µm-sections were mounted on gelatin-coated slides dewaxed twice in xylene and rehydrated by a series of decreasing ethanol concentrations. The histological evaluation was performed after staining the sections with a standard hematoxylin/eosin-based method (0.1% H&E).

### 2.5. Analysis of Hydroxyproline (Hyp) in Liver Tissue

The spectrophotometric analysis for Hyp levels in liver was used as an indirect measure of tissue collagen content and to quantify the liver damage by a Woessner method [28] with slight modifications.

### 2.6. Determination of Glutathione S-Transferase (GST)

The glutathione-S-transferase activity (GST) in liver homogenates was determined using a 1-chloro-2,4-dinitrobenzene reagent [29]. The chemical mixture included 0.1 M phosphate buffer (pH = 8.2), 15 mM GSH, and 15 mM CDNB which are added to a 10 µL tissue homogenate sample. A control sample contained only a chemical reaction mixture without tissue homogenate. The enzyme production and, respectively, the activity of GST in each investigated sample, were evaluated through the spectrophotometric sample investigation with absorption at 340 nm.

### 2.7. Experimental Protocol for Functional Markers Measurement of Hepatic Damage

For evaluation of the functional hepatic damage biomarkers [30,31], the aspartate aminotransferase (AST), alanine aminotransferase (ALT), and alkaline phosphatase (ALP) levels were tracked. The serum fractions were collected after centrifugation of blood samples at 4000 rpm for 10 min (4 °C). A quantified assessment of the biochemical parameters AST, ALT, and ALP levels were made by commercial enzymatic kits (Sigma-Aldrich Pty Ltd., Merck KGaA, Sofia, Bulgaria).

### 2.8. Lipid Peroxidation Analysis

Total ROS overproduction in liver tissue and serum was examined by the EPR method (X-Band, Emxmicro, Bruker) according to the method by Shi et al. [32].

### 2.9. NO Radical Formations

The •NO radical formations in liver homogenate and serum were evaluated by EPR methods of Yoshioka et al. [33] and Yokoyama et al. [34].

### 2.10. 3-Maleimido proxyl (5-MSL) Protein Oxidation in the Liver

The degree of protein/albumin damage in liver tissue was assessed by the EPR method using spin-conjugation with 3-maleimido proxyl (5-MSL), according to Takeshita et.al. [35].

All EPR measurements were in arbitrary units (a.u.).

### 2.11. Immunoenzyme Assays

#### 2.11.1. Determination of the Activity of Antioxidant Enzyme System and Products of Oxidation of Proteins and Lipids in the Liver

To determine the enzyme activity of superoxide dismutase (SOD) and glutathione peroxidase (GSPx), and oxidative stress parameters malondialdehyde (MDA), protein carbonyl content (PCC) and glycation end products (AGEs) were used ELISA methods.

#### 2.11.2. Measurement of Proinflammatory Markers in Liver Tissue and Blood

The proteins (pro-collagen I alpha 1 (PGC-1) and gene SIRT1) and cytokine levels (IL-1β, IL-10, IL-6, ITF-γ, and TNF-α) were measured with ELISA kits on the 14th day and on the 28th day from the beginning of the experiment.

### 2.12. Statistical Analysis

The obtained results were processed on Statistic 8.0 (Stasoft, Inc., Tulsa, OK, USA), to determine statistically significant differences between groups. Student-*t*-tests were used. Results were expressed as mean ± standard error (SE), and *p* < 0.05 was considered statistically significant.

## 3. Results

### 3.1. Physiological Status

Mice were weighed daily during the experimental period of 28 days; their general physical condition and food intake were also monitored. Table 1 shows the changes in body weight in OTA-induced hepatotoxicity and in the other three groups measured on the 5th, 8th, 14th and 28th day.

### 3.2. Liver Histopathology

The liver tissue examination from the three groups of experimental animals showed varying degrees of morphological changes. We used our own protocol to account for pathomorphological changes in different groups according to their severity (Table 2).

Mild degenerative changes characteristic of the first minutes and hours after death were observed in the liver tissue of the control group and group protected with *A. indica* oil. The changes in the group treated with OTA alone, as well as in the group treated with a combination of *A. indica* oil and OTA, are significantly more pronounced, with degenerative, necrotic, inflammatory, and vascular changes.

In animals treated with OTA alone, more pronounced inflammatory changes were observed, consisting of pronounced infiltration of mononuclear inflammatory cells and proliferation of Kupffer cells, these changes being more pronounced here than in the group treated with the combination *A. indica* oil and OTA. In addition, more pronounced vascular congestion was reported, as well as areas of degenerative and necrotic changes. Regarding nuclear changes in hepatocytes, only the group with OTA showed such in isolated cases (Figure 1).

### 3.3. Functional Markers of Hepatic Damage

The mice treated with OTA alone compared to the control group had statistically significant increased serum levels of hepatic enzymes AST (364.5 ± 16 (U/L) vs. 219.9 ± 10 (U/L), *p* < 0.05, Figure 2A), ALT (86.2 ± 1.1 (U/L) vs. 26.1 ± 0.7 (U/L), *p* < 0.05, Figure 2B), and ALP (69.4 ± (U/L) vs. 31.7 ± 3.5 (U/L), *p* < 0.05, Figure 2C), confirming hepatic cell injury. The enzyme levels in the group treated with OTA alone were statistically significantly higher compared to the group protected with *A. indica* oil + OTA, AST (364.5 ± 16 (U/L) vs. 242.7 ± 8.6 (U/L), *p* < 0.05, Figure 2A), ALT (86.2 ± 1.1 (U/L) vs. 45.9 ± 0.6 (U/L), *p* < 0.05, Figure 2B), and ALP (69.4 ± (U/L) vs. 51.4 ± 2.1 (U/L), *p* < 0.05 (Figure 2C).

Statistically significant differences were also observed in the results of the OTA group compared to the group protected only with *A. indica* oil: AST (364.5 ± 16 (U/L) vs. 216.7 ± 5.9 (U/L), *p* < 0.05, Figure 2A), ALT (86.2 ± 1.1 (U/L) vs. 24.8 ± 0.8 (U/L), *p* < 0.05, Figure 2B), and ALP (69.4 ± (U/L) vs. 32.1 ± 3.5 (U/L), *p* < 0.05, Figure 2C). Furthermore, in the group treated with a combination of *A. indica* and OTA, the levels of hepatic enzymes were effectively reduced and showed hepatoprotective properties. The hepatoprotective properties of *A. indica* oil were confirmed in Figure 1 and Table 1. Figure 1a–h showed no significant pathological changes in the liver samples. Table 1 reported no drastic differences in BW between the control group and mice treated with *A. indica* oil alone and in combination with *A. indica* oil and OTA.

### 3.4. The Effects on SIRT1/PGC-1α Pathway

SIRT1 mediates lipogenesis checkpoint responses and directly inhibits ROS/RNS generation (Figure 3A). The SIRT1-mediated PGC-1α pathway plays a key role in regulating the antioxidant status in OTA-injured mice. In this case, the extent PGC-1α and SIRT1 are involved in liver damage resulting from OTA intoxication was investigated. Figure 3A,B shows that the protein expressions of SIRT1 and PGC-1α in the OTA group significantly decreased (*p* < 0.05).

After protection with *A. indica* oil alone and in combination with *A. indica* oil and OTA, an increase in levels of SIRT1 and PGC-1α was observed compared to the group treated with OTA alone on both day 14 and day 28 (*p* < 0.001). The group treated only with *A. indica* oil had SIRT1 (Figure 3A) and PGC-1α (Figure 3B) statistically insignificant differences to the control (*p* > 0.05).

### 3.5. Analysis of Hydroxyproline (Hyp) and MDA in Liver Tissue

Figure 4 presents the OTA toxicity by measuring the hydroxyproline content in liver tissue and lipid peroxidation levels measured as MDA substances.

OTA long-term administration statistically significantly increased the Hyp levels vs. controls (849.18 ± 108.4 mg/g tissue vs. 405.56 ± 74.8 mg/g tissue, *p* < 0.05). The results presented in Figure 4A show that OTA-induced liver damage was compensated by *A. indica* oil protection. The Hyp level was statistically significantly reduced in the group treated with the combination of *A. indica* and OTA (849.18 ± 108.4 mg/g tissue vs. 549.09 ± 81.9 mg/g tissue, *p* < 0.05). The group protected for 28 days with *A. indica* oil only showed lower levels with no statistically significant difference compared to controls (Figure 4A).

In OTA-treated mice, the MDA levels were significantly increased in the liver compared to controls (4.55 ± 0.8 µmol/mL vs. 2.16 ± 0.44 µmol/mL, *p* < 0.05, Figure 4B). Pre-treatment with *A. indica* oil effectively brought down liver MDA levels compared to the OTA-treated group (2.74 ± 0.57 µmol/mL vs. 4.55 ± 0.8 µmol/mL, *p* < 0.05, Figure 4B) and close to the control (2.74 ± 0.57 µmol/mL vs. 2.16 ± 0.44 µmol/mL, *p* > 0.05, Figure 4B). The results mean that *A. indica* oil reduced the OTA toxicity and possesses hepatoprotective properties.

### 3.6. Effect of Antioxidant Enzymes

GST activity (Figure 5A) was statistically significantly increased in the liver of OTA-injured mice compared to the control (692.12 ± 55.14 nmol/gPr vs. 502.3 ± 58.59 nmol/gPr, *p* < 0.05) vs. the protected with *A. indica* oil group (692.12 ± 55.14 nmol/gPr vs. 509.4 ± 62.78 nmol/gPr, *p* < 0.05). The group treated with a combination of *A. indica* oil and OTA showed a statistically insignificant decrement in GST activity compared to the OTA group (601.8 ± 59.14 nmol/gPr vs. 692.12 ± 55.14 nmol/gPr, *p* > 0.05, Figure 5A).

The SOD activity (Figure 5B) in the OTA group was significantly decreased compared to the control after 28-day exposure (1.3 ± 0.02 U/gPr vs. 2.44 ± 0.07 U/gPr, *p* < 0.05), whereas in the group with a combination of *A. indica* oil protection and OTA, the levels of SOD activity were markedly elevated (1.3 ± 0.02 U/gPr vs. 1.9 ± 0.03 U/gPr, *p* < 0.05). SOD activity in the group protected with *A. indica oil* only was similar to the controls (2.47 ±0.04 U/gPr vs. 2.44 ± 0.07 U/gPr, *p* < 0.05, Figure 5B).

Statistically significant lower GSPx liver activity was measured in mice treated with OTA compared to the healthy controls (20.31 ± 1.2 U/gPr vs. 66.14 ± 3.44 U/gPr, *p* < 0.05, Figure 5C), and to the group treated with a combination of *A. indica* oil and OTA (20.31 ± 1.2 U/gPr vs. 51.35 ± 4.14 U/gPr, *p* < 0.05). The liver GSPx activity in *A. indica*-protected mice were significantly elevated compared to the OTA group (68.48 ± 3.07 U/gPr vs. 20.31 ± 1.2 U/gPr, *p* < 0.05).

No significant changes in the activity of SOD, GSPx, and GST were indicated between the mice treated with *A. indica* alone and the control group.

### 3.7. Determination of Oxidative Protein Remodeling in Liver Tissue

Liver tissue is highly vulnerable to oxidative injury and OTA-induced mycotoxicity. The AGEs levels (Figure 6A) were statistically significantly increased in the OTA-treated group compared to the healthy group (802.59 ± 23.47 mg/mL vs. 243.35 ± 21.99 mg/mL, *p* < 0.05). A statistically insignificant increase compared to the controls was observed in the group protected with *A. indica* oil only (251.76 ± 20.19 mg/mL vs. 243.35 ± 21.99 mg/mL, *p* > 0.05) and in the group treated with a combination of *A. indica* oil + OTA (359.7 ± 18.86 mg/mL vs. 243.35 ± 21.99 mg/mL, *p* > 0.05). A statistically significant reduction compared to the OTA group in AGEs levels was detected in *A. indica* oil only (251.76 ± 20.19 mg/mL vs. 802.59 ± 23.47 mg/mL *p* < 0.05) and the group treated with the combination *A. indica* + OTA (359.7 ± 18.86 mg/mL vs. 802.59 ± 23.47 mg/mL, *p* < 0.05, *t*-test).

The liver homogenate expression of 5-MSL conjugated proteins (Figure 6B) was statistically significantly higher after controlled OTA administration compared to the control group (*p* < 0.05). The *A. indica* oil significantly reduced hepatic protein up-regulation in the liver of the OTA-treated group (*p* < 0.05). Nitroxides were spread to the liver tissue and used for the evaluation of conformational changes in albumin [35]. Reading available oxidative proteins remodeling is assessed by EPR [36], with a 5-MSL spin-trap.

The PCC levels (Figure 6C) in the group treated with OTA were statistically significantly increased compared to healthy mice (10.89 ± 1.12 nmol/mL vs. 5.02 ± 0.81 nmol/mL, *p* < 0.05). A statistically significant difference was observed in the OTA-treated group compared to the protected group with the combination *A. indica* oil and OTA (10.89 ± 1.12 nmol/mL vs. 6.23 ± 0.75 nmol/mL *p* < 0.05) and the group treated only with *A. indica* oil (10.89 ± 1.12 nmol/mL vs. 4.78 ± 0.5 nmol/mL *p* < 0.05).

### 3.8. Parameters of Oxidative Damage in Liver Tissue

As a result of the significant depletion in the activity of the antioxidant enzymes SOD and GSH, as well as Hyp and GST, the EPR method showed the overproduction of free radicals in both liver homogenate and blood serum. The statistically significant increase in the NO radical levels (Figure 7A) and ROS levels (Figure 7C) measured in the liver is due to OTA exposure (*p* < 0.05). A statistically significant reduction in NO radical level was measured in the group treated with the combination *A. indica* oil+ OTA compared to the OTA group (*p* < 0.05) in serum (Figure 7B). Moreover, the values registered in the group with combined therapy are close to the control group and the group treated with *A. indica oil*, only. The decrease in ROS levels measured in serum indicated its restorative effect against OTA-induced damage in the oil and OTA-protected group, compared to the OTA-only group (*p* < 0.05) (Figure 7D).

### 3.9. IL-1b, IL-6, IL-10 in Liver Homogenate/Serum of OTA-Injured Group, Compare to Group Protected with A. indica Oil Alone, Group with Combination Oil + OTA and Control

Compared to the healthy mice, OTA treatment increased interleukins levels in liver homogenate IL-1β (187.5 ± 12.5 pg/mL vs. 144.7 ± 10.01 pg/mL, *p* < 0.05), IL-10 (9.2 ± 0.8 vs. 7.1 ± 0.9; *p* < 0.05), IL-6 (176.6 ± 15.8 pg/mL vs. 145.2 ± 21 pg/mL; *p* < 0.05) and serum IL-1β (178.2 ± 21.8 pg/mL vs. 140.5 ± 15.67 pg/mL, *p* < 0.05), IL-10 (212.6 ± 17.55 vs. 186.4 ± 2.8; *p* < 0.05), IL-6 (179.2 ± 21.3 pg/mL vs. 142.8 ± 27.1 pg/mL; *p* < 0.05), (Figure 8A–F). The group protected with *A. indica* oil alone and the combination of *A. indica* + OTA (14 days) and (28 days) showed results similar to the controls. In addition, the oil-protected groups (alone and in combination with OTA) showed a significant decrease in interleukin levels compared to the OTA group in the liver homogenate (*p* < 0.05) and serum (*p* < 0.05).

### 3.10. TNF-a and INF-γ Concentration of A. indica Oil Protected Group Treated with OTA, with A. indica oil Alone and OTA-Damaged Group Compared to Healthy Mice

OTA administration increased the concentrations of TNF-α (12.9 ± 1.2 pg/mL vs. 9.01 ± 0.12 pg/mL in liver homogenate, (*p* > 0.05) and 79.45 ± 27.01 pg/mL vs. 63.24 ± 14.77 pg/mL in serum (*p* > 0.05)), and INF-γ (10.71 ± 0.05 pg/mL vs. 8.2 ± 0.07 *p* > 0.05 in liver homogenate (*p* > 0.05) and serum 189.8 ± 24.88 pg/mL vs. 145.7± 26.14 pg/mL, *p* > 0.05) compared to the control in liver homogenate and serum on the 28-day course (Figure 9A–D).

*A. indica* oil hepatoprotection markedly decreased the concentration of TNF-α (*p* < 0.05) and INF-γ (*p* > 0.05) compared to the OTA-induced toxicity group (*p* < 0.05) on the 28-day course, respectively. The *A. indica* oil therapy (120 mg/mL) insignificantly lowered the serum INF-α and TNF-γ expression on the 14^th^ day and the last 28th day (Figure 9B,D), compared to OTA-administration. The OTA-induced liver and serum modifications in the concentration of TNF-γ (*p* > 0.05; Figure 9A,B) and INF-α (*p* > 0.05; Figure 9C,D) were significantly attenuated with *A. indica* protection on day 28 (*p* < 0.05). Moreover, the group treated with *A. indica* oil showed protection from serum and liver inflammations and the results were almost analogous to the controls.

### 3.11. Correlations between Parameters

In an analysis of the results of the markers examined in this study, a positive correlation was found in PCC vs. AGEs (r = 0.96, *p* < 0.05); PCC vs. MDA (r = 0.91, *p* < 0.05); PCC vs. 5-MSL (r = 0.91, *p* < 0.05); MDA vs. AGEs (r = 0.92, *p* < 0.05); MDA vs. NO radicals (r = 0.89, *p* < 0.05); MDA vs. ROS (r = 0.94, *p* < 0.05); GSH vs. ROS (r = 0.91, *p* < 0.05); GSH vs. SIRT1 (r = 0.91, *p* < 0.05); GSH vs. PGC-1α (r = 0.91, *p* < 0.05); GSH vs. 5-MSL (r = 0.92, *p* < 0.05), IL-1β vs. TNF-γ (r = 0.81, *p* < 0.05); and PCC vs. IL-6 (r = 0.85, *p* < 0.05).

## 4. Discussion

The analysis of the results presented in this study shows that the group treated with OTA showed increased oxidative stress by depleting the activity of antioxidant enzymes and showing a drastic increase in ROS/RNS levels, leading to protein damage and increased levels of inflammatory cytokines. Cells quality control systems that the homeostasis of oxidative stress were reduced in mycotoxicosis due to an increase in inflammatory cytokines, leading to the accumulation of oxidized biomolecules [37]. Damaged proteins and lipids accumulated in stressed cells alter the structural and potential function of protein adducts, leading to changes in cellular metabolism and functioning in a state of mycotoxicosis [38]. According to Tao et al. [39], OTA stimulates both NADPH-dependent and ascorbate-dependent lipid peroxidation with co-factor ferrous ions (Fe^3+^). Therefore, the OTA-Fe^3+^ complex in the presence of NADPH-cytochrome P450 reductase system facilitates the reduction in ferric ions to iron ions, and the resulting OTA-Fe^2+^ complex generates •OH, which increases MDA and ROS levels [40]. In addition, plasma membrane permeability and calcium homeostasis are impaired, affecting calcium-sensitive channels and leading to increased ROS and RNS free radical initiation [41]. The involvement of ROS/RNS in the development of oxidative stress due to mycotoxicosis is a proven fact, and the presented results show that the intensity of pathomorphological and biochemical changes and deterioration of oxidative status is best expressed in the group treated with OTA [42]. 

Cells have various mechanisms to prevent ROS-induced damage, which include enzyme systems (SOD, CAT), glutathione liver enzymes, and endogenous antioxidant molecules such as glutathione (GSH). Glutathione is the most important antioxidant synthesized in cells and helps to remove peroxides and many xenobiotic compounds [43]. In addition, GSH is a cofactor and substrate for glutaredoxins (Grx), which catalyzes disulfide reductions in the presence of NADPH and glutathione reductase GR. Protein and lipid damage is prevented by enzymes such as GR, multiple glutaredoxins (GSPx), and glutathione-S-transferases (GST). For example, the antioxidant enzyme SOD degrades the superoxide anion radical (•O_2_^−^) to H_2_O and O_2_, while GPx reduces H_2_O_2_ to water and GSH, and the glutathione reductase GR regenerates GSH [44]. The antioxidant protection mechanism against peroxides involves glutathione, which is involved in the physiological control of free radical generation and increased oxidative stress through the interaction and expression of GR, GPx, and GST [44,45]. Glutathione reductase (GR) is a homodimeric enzyme that catalyzes the oxidized glutathione (GSSG) reduction process to GSH, using FAD and NADPH as cofactors, thus participating in the maintenance of redox balance in living organisms and helping to remove peroxides and many xenobiotic compounds. Glutathione peroxidases (GPXs) are differentially expressed in cells and tissues and are involved in H_2_O_2_ detoxification; members of this family reduce peroxides produced by peroxisomes and mitochondria in cells in the cytosol. The activity of glutathione-S-transferases (GST) depends on the presence of GSH, whose role is to detoxify reactive electrophilic compounds, including toxins and products of oxidative stress. The mechanism involves the conjugation of free radicals and non-radical molecules with reduced glutathione GSH conjugated products, which are removed from the cell by specific transporters [46]. OTA induces ROS, thereby modulating the inflammatory response by up-regulating pro-inflammatory cytokines, down-regulating anti-inflammatory IL-4 expression, cytochrome P450 activity, and increasing polyunsaturated fatty acid metabolism [45]. Furthermore, OTA mycotoxicity depletes antioxidant enzymes (GSPx and SOD) and GST at the cellular level and increases the 8-oxoguanine formation and protein carbonylation [47]. In addition, a dose-dependent increase in ROS/RNS due to OTA intoxication in primary proximal tubular cells of experimental animals and cell lines leads to the depletion of intracellular glutathione and cell death [48]. The increased MDA and ROS/RNS in the OTA group suggested that the damage caused was related to impaired oxidative pathways [48]. The presented results reported significant changes in antioxidant enzyme activity and hydroxyproline concentration in the OTA (1.25 mg/kg)-treated group, suggesting oxidative tissue damage and impaired antioxidant status. The elevated liver enzymes (AST, ALP, and ALT) indicate increased oxidative stress and OTA toxicity [48]. The high ROS and RNS led to a significant increase in peroxidase and enlarged phenoxyl radical production [49]. Furthermore, under the GSH action, the phenoxyl radicals were converted back to reactive species, which increased the formation of •O_2_^−^ as a by-product. This led to the depletion of the activity of the antioxidant enzymes (SOD, GSRx) and GST and oxidative damage to biomacromolecules (lipids and proteins).

The *A. indica* oil possesses a phytoremediation capability and acts as a protector that directly inhibits high ROS and RNS, regulates protein oxidation, collagen deposition, and reduces mycotoxicity OTA [44,45]. Analysis of the results in the group protected with the combination of *A. indica* and OTA showed the hepatoprotective role of the oil [50]. These results are consistent with studies reporting the protective effect of *A. indica* oil against induced liver damage after intoxication with CCl_4_ and Cisplatin [51,52]. Yanpallewar et al. [53] reported liver damage in rats after paracetamol poisoning. *A. indica* shows high hepatoprotection, demonstrated by stabilizing the serum activity of liver enzymes ALP, ALT, AST, and histopathological observations of liver tissue. Bhanwra [54] reported protection by *A. indica* leaf extract in paracetamol-damaged rats, resulting in normal liver appearance and histology, significantly decreased MDA and increased antioxidant and detoxifying enzymes. The oil isolated from *A. indica* seeds contains active ingredients and secondary metabolites, which have antioxidant, antifungal, antiviral, and anti-inflammatory properties [55,56].

According to Gossé et al. [57] *A. indica* oil is rich in polyunsaturated fatty acids (PUFA) and improves antioxidant protection by significantly increasing the body’s activity against viruses and mycotoxins. The PUFAs act directly or through their metabolites on signaling pathways and affect many metabolites, reducing TBARS production, superoxide anion secretion, and LDL peroxidation [58]. Based on these data, we treated a group of mice (*n* = 6) with *A. indica* seed oil (170 mg/kg), both alone and in combination with OTA (1.25 mg/kg) and monitored the protective effect of the oil against mycotoxicosis. The analysis of the results shows that *A. indica* seed oil in combination with OTA has a strong hepatoprotective and antioxidant effect. The activities of serum liver enzymes AST, ALP, and ALT were statistically significantly reduced (Figure 3), while SIRT1 (Figure 4A) and PGC-1 (Figure 4B), Hydroxyproline (Figure 5A), MDA (Figure 5B), and the activity of antioxidant enzymes GSPx, SOD and GST were increased (Figure 6A–C).

In the group protected with *A. indica* oil, no degenerative changes in the liver were observed (Figure 1). The cell depletion in the immunocompetent organs of experimental animals treated with a combination of *A. indica* oil and the OTA showed oil immunostimulatory, hepatoprotective, and immunoprotective properties. Baligar et al. reported a hepatoprotective role of *A. indica* and its active constituents in hepatotoxicity in rats [59]. Histology and ultrastructure results confirmed that *A. indica* dose-dependent treatment reduced hepatocellular necrosis and moderately restored rat liver to normal [59]. The study investigating the protective effect of *A. indica* oil against CCl4-induced hepatotoxicity in rats demonstrated a hepatoprotective effect of the *A. indica* oil with an efficacy close to that of silymarin [60].

Our 28-day protection with *A. indica* oil in OTA mycotoxin food-fed mice reported a significant decrease in hepatocyte density, and the results were close to the controls. Moreover, the combined use of *A. indica* oil and OTA statistically significantly reduced mycotoxicosis, extracellular matrix, and protein modification. Therefore, this led to an improvement in the oxidative status of the oil-treated mice compared to the OTA-only treated mice. Protecting with *A. indica* oil has a beneficial effect on OTA-induced hepatotoxic processes and probably participates in the re-modulation of the Th2 immune response. Furthermore, damaged liver tissue and increased levels of inflammatory enzymes resulting from the development of protein dysfunction induce carbonyl stress due to OTA mycotoxicosis [48,60,61].

SIRT1 is correlated with mitochondrial function, energy metabolism, autophagy, apoptosis, and oxidative stress processes and plays a significant role in liver diseases. SIRT1 overexpression leads to increased levels of PPARα and its coactivator PPARγ by a mechanism including acetylation of the peroxisome proliferator-activated receptor-γ coactivator-1α, PGC-1α, which can regulate a function of the multi-functional transcriptional coactivator [62]. In the present study, we analyzed the gene and protein expressions of SIRT1 and PGC-1α in the liver. The results showed that OTA significantly down-regulated the protein levels of SIRT1 and PGC-1α, which suggested that inhibition of the SIRT1/PGC-1α pathway will impair the quality of mitochondrial synthesis, thereby exacerbating oxidative stress. Activating the transcription of PGC-1α is related to enhancing tissue antioxidant capacity of SIRT1 and the expression of superoxide [23,24], and the SIRT1/PGC-1α cascade may act as an activator of the Nrf2 signaling pathway and reduce OTA-oxidative injury. Activated Nrf2 provoked an antioxidant response, resulting in the activation of the endogenous antioxidant enzymes [23,24]. *A. indica* showed a positive effect on OTA-induced hepatic injury by increasing the levels of SIRT1 and PPAR-α. Therefore, we hypothesize that pretreatment of mice with *A. indica* attenuates OTA-induced liver injury by expressing SIRT1, stimulating PGC-1α production, increasing essential fatty acid levels, reducing lipogenesis, and improving lipid metabolism.

Moreover, the mice treated with OTA, the AGEs products (Figure 7A) and carbonylated proteins (Figure 7B) were statistically significantly increased, and the reduction in nitroxide protein (Figure 7B) was almost twice delayed and liver damage was registered. Protection with *A. indica* oil in a combination treatment restores these parameters almost to control levels. Therefore, *A. indica* oil modulates OTA-induced inflammation and progressive hepatocyte processes in the liver by lowering PCC, reducing protein dysfunction, and completely inhibiting the concentrations of •O_2_ˉ, •OH, NO• radicals. Furthermore, *A. indica* oil rapidly inhibits AGEs glycation and the action of proteins and thiols [63]. The active compounds from *A. indica* deactivated the protein and lipid oxidation and suppressed procoagulant activity in hepatic cells, which equilibrated collagen turnover during an exchange in antioxidant-prooxidant balance [52]. The spin-labeled 5-MSL scavenged membrane proteins alkylated as a result of the reaction between the maleimide group with the SH sites, i.e., by rapid rotation, giving an isotropic triplet spectrum with sharp lines (not shown); nitroxyl captures the weakened SH regions of the corresponding amino acid [35] and due to limitations in movement, the EPR spectrum changes shape as a result of anisotropy [36]. Long-term protection with *A. indica* (170 mg/kg) against OTA-induced mycotoxicosis completely remodulates the concentration of 5-MSL, reduces membrane proteins, and probably further reduces the levels of protein aggregates and, thus, restores antioxidant status to levels close to the controls.

In the OTA-induced mycotoxicosis model, ROS/RNS generation resulted in progressive lipid peroxidation that affected the reductive protein metabolism, increased collagen synthesis by increasing hydroxyproline and directly affected the antioxidant defense system. OTA leads to oxidation, and oxidation reactions occur through the cytochrome protein family. Among the isoforms of CYP450 enzymes, cytochrome expression is relatively high in the liver. When toxic substances enter the body, transcription factors migrate from the cytoplasm to the nucleus and increase the expression of cytochrome proteins, leading to oxidation. Antioxidant enzymes are involved in the process to provide protection against oxidative stress [63].

An anti-inflammatory IL-10 is cytokine produced mainly by macrophages and Th2 cells. The reported biological activities of IL-10 that may be interrelated include the inhibition of macrophage-mediated cytokine synthesis, inhibition of delayed-type hypersensitivity reactions, and stimulation of the Th2 cellular response, leading to increased antibody production [63,64]. IL-10 functions by inhibiting proinflammatory cytokines produced by macrophages and regulatory T cells, including IFN-gamma and TNF-alpha. In the present study, *A. indica* was shown to significantly inhibit OTA-induced levels of TNF-α, IL-1β, IL-6, IL-10, and IFN-gamma in mice. Rodrigues et al. [65] reported that the suppressive inflammatory effects of *A. indica* extract completely inhibited OTA production. Schumacher et al. [66] reported that *A. indica* leaf extract induced apoptosis in leukemia cells and inhibited proliferation in prostate cancer. The leaf extract also stimulates immune function by activating killer cells and has an antiproliferative effect, which is probably due to the active compound in *A. indica* oil.

## 5. Conclusions

In the present study, we reported significantly elevated cytocines, interferon-gamma (IFN-γ), tumor necrosis factor-alpha (TNF-α), and collagen synthesis in the OTA-treated mice. Protection with *A. indica* oil (170 mg/mL) induced cell proliferation of normal hepatocytes and a probable anti-inflammatory therapeutic effect. The results presented above suggest inhibition of OTA-induced inflammation by *A. indica* oil and ROS reduction influx by regulating IL-1β, IL-6, IL-10, IFN-γ, and TNF-α balance after a 28-day course. OTA is a mycotoxin with a hepatotoxic effect, and toxicity resulted in statistically significant increases in biomarkers of oxidative stress, abnormal hepatic and antioxidant enzymes, and cytokine levels. *A. indica* oil has a potential restorative and protective effect, as a result of which hepatic and antioxidant enzymes are restored. In addition, the protective effects of *A. indica* seed oil have led to functional recovery in the OTA-induced mice model. Therefore, *A. indica* oil can be used as a practical approach to safe OTA-contaminated feed use.

## Data Availability

Not applicable.

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
