# Peer review of "The Azadirachta indica (Neem) Seed Oil Reduced Chronic Redox-Homeostasis Imbalance in a Mice Experimental Model on Ochratoxine A-Induced Hepatotoxicity"

_antioxidants, 2022, doi:10.3390/antiox11091678_

Round 1

Reviewer 1 Report

Durung the reading of the manuscript I have found several places hard to read and follow due to the sometimes strange convolution of sentences. I strongly suggest throrough English editing. I wish in my studies the interindividual differences between the 6 animals were so small as here. As no raw data are provided e.g. as a supplement, we can only admire small error bars. My comments and points are presented below in the order of appearance.

line 50. The sentence beginning with „Although, …” needs to be revised. Add to the sentence by whom it was discussed, add name of the author of the paper (12).

line 51. typo: hepatotoxicity

line 55, “in particular” as an injection should be flanked with commas

line 56. use full name before the abbreviation “Gap junction intercellular communication (GJIC)”

line 58. OTA is not being damaged, e.g. “damage due to OTA”

line 66 how A.indica influences DNA replication?, in current state, the sentence makes little sense

line 68. starting from this line correct all A. indica names to be written with cursive

line 81. change junction of the sentences to be like “and impairs synthesis of prostaglandins”

line 91. use the full name at the first site of usage:  peroxisome proliferator-activated receptor gamma coactivator 1-α instead of PGC-1α

line 115. the scientific name of the plant in italics

table 1. should be “control”

line 178. why do you use the term in vivo (in alive) for measurements with the EPR of pieces of tissue which must have been placed in some kind of vessel? In vivo implies maintaining animals alive, and performing measurements on their liver intact. Hence more proper would be in vitro (in glass tube)

line 229. Crucial point of the review: The description of the histopathology results is missing microphotographs and the description of liver from animals on the diet supplemented with  A. indica oil only.  Include these results.

line 312. what do you mean by p<0.00 in T-test?

line 323 why p=0.00 means lack of difference between A.indica oil pre-treated and control animals?

line 332, line 338, lines 380-420 please clarify and correct p-value going into negative (“p<0.00”), into appropriate threshold value for assigning significance

line 386, the term “therapy” is not adequate here. The mice were not treated to cure a disease. Pleas change the wording

line 427-430. The sentences here are grammatically not correct and making your reasoning  very hard to follow.

line 435 change “iron ions” into “ferrous ions”, as the meaning is Fe2+ (II)

line 452-453 Please support your claim “GSH (...) expresses enzymes with antioxidant activity to control.”) and provide a proper reference to the source.

line 448-451 I can’t agree with the description of the function of GST presented here. The reference describes an anti-oxidative action of plant enzymes, whereas mammalian liver GST would be much more fitting. Moreover, the article cited does not use at all words “GST” or “glutathione-S-transferase” .

line 477. logical error in the sentence “The A. indica oil used in the study is not only a phytoremediation plant (...).”

line 486. use “damaged” instead of “induced”

line 511. The sentence starting with “After protection with ...” should be edited, maybe re-made into two or three shorter sentences. It’s hard to understand your conclusions.

Conclusions: 1) Improve language of the manuscript. 2) Present photographs of tissues from animals treated only with the neem oil.

Author Response

Dear reviewer,

Thank you very much for helping us to do our manuscript better.

In the manuscript text, the corrections for your questions are highlighted in yellow.

Durung the reading of the manuscript I have found several places hard to read and follow due to the sometimes strange convolution of sentences. I strongly suggest throrough English editing. I wish in my studies the interindividual differences between the 6 animals were so small as here. As no raw data are provided e.g. as a supplement, we can only admire small error bars. My comments and points are presented below in the order of appearance.

Point 1: line 50. The sentence beginning with „Although, …” needs to be revised. Add to the sentence by whom it was discussed, add name of the author of the paper (12).

Response 1: Imaoka et al., [12] reported acute hepatotoxicity induced by OTA and concluded that OTA metabolism varies among different tissues.

Point 2: line 51. typo: hepatotoxicity

Response 2: done, we corrected 

Point 3: line 55, “in particular” as an injection should be flanked with commas

Response 3: OTA-induced acute liver injury contributes to the weakening of the body's ability to counteract the redox-homeostatic imbalance and, in particular,

Point 4: line 56. use full name before the abbreviation “Gap junction intercellular communication (GJIC)”

Response 4: to stop altered gap junctional intercellular communication (GJIC).

Point 5: line 58. OTA is not being damaged, e.g. “damage due to OTA”

Response 5: In addition, chronic exposure to OTA, leads to hepatocarcinoma development….

Point 6: line 66 how A.indica influences DNA replication?, in current state, the sentence makes little sense

Response 6:  A. indica oil has a high percentage of antioxidant activity due to its total phenolic content and great free radical scavenger ability

Point 7: line 68. starting from this line correct all A. indica names to be written with cursive

Response 7:  done

Point 8: line 81. change junction of the sentences to be like “and impairs synthesis of prostaglandins”

Response 8:  and impairs the synthesis of prostaglandins, leukotrienes

Point 9: line 91. use the full name at the first site of usage:  peroxisome proliferator-activated receptor gamma coactivator 1-α instead of PGC-

9: SIRT1 functions together with peroxisome proliferator-activated receptor gamma coactivator 1-α (PGC-1α) and encode the nicotinamide adenine dinucleotide (NAD+)-dependent histone deacetylase.

Point 10: line 115. the scientific name of the plant in italics

Response 10: Done

Point 11: table 1. should be “control”

Response 11: Done

Point 12: line 178. why do you use the term in vivo (in alive) for measurements with the EPR of pieces of tissue which must have been placed in some kind of vessel? In vivo implies maintaining animals alive, and performing measurements on their liver intact. Hence more proper would be in vitro (in glass tube)

Response 12: Done

Point 13: line 229. Crucial point of the review: The description of the histopathology results is missing microphotographs and the description of liver from animals on the diet supplemented with  A. indica oil only.  Include these results.

Response 13: In Figure 1 and Table 2 we add the histopathology results from group mice protected with A. indica only.

Point 14: line 312. what do you mean by p<0.00 in T-test?

Response 14: Thank you, my mistake we corrected:

The mice treated with OTA alone compared to the control group had statistically significant increased serum levels of hepatic enzymes AST (364,5 ± 16 (U/L) vs 219,9 ± 10 (U/L), p<0, 05, Figure 2, A), ALT (86,2 ± 1,1 (U/L) vs 26,1 ± 0,7 (U/L), p<0, 05, Figure 2 B), and ALP (69,4 ± (U/L) vs 31,7 ± 3,5 (U/L), p<0,05, Figure 2, C), which was hepatic cell injury confirmation. The enzyme levels in the group treated with OTA alone were statistically significantly higher compared to the group protected with A. indica oil + OTA, AST (364,5 ± 16 (U/L) vs 242,7 ± 8,6 (U/L), p<0,05, Figure 2, A), ALT (86,2 ± 1,1 (U/L) vs 45,9 ± 0,6 (U/L), p<0,05, Figure 2 B), and ALP (69,4 ± (U/L) vs 51,4 ± 2,1 (U/L), p<0,05, Figure 2, C).

Point 15: line 323 why p=0.00 means lack of difference between A.indica oil pre-treated and control animals?

Response 15: The protected with A. indica oil alone group IL-1β (138,2 ± 30,41 pg/ml vs 144,7 ± 18,07 pg/ml), IL-10 (7,2 ± 0.98 pg/ml vs 7,1 ±0.5 pg/ml), IL-6 (139,5 ± 12.77 pg/ml vs 145,2± 15.08 pg/ml) in liver homogenate and in serum IL-1β (146,2 ± 21,5 pg/ml vs 140,5 ± 10,9 pg/ml), IL-10 (191,3 ± 21,25 pg/ml vs 186,4 ± 19,04 pg/ml), IL-6 (145,6 ± 15.13 pg/ml vs 142,8 ± 13, 18 pg/ml) and combination of A. indica + OTA (14 days) and (28 days) showed results similar to controls.

Point 16: line 332, line 338, lines 380-420 please clarify and correct p-value going into negative (“p<0.00”), into appropriate threshold value for assigning significance

Response 16: Done

Point 17: line 386, the term “therapy” is not adequate here. The mice were not treated to cure a disease. Pleas change the wording

Response 17: Done

Point 18: line 427-430. The sentences here are grammatically not correct and making your reasoning very hard to follow.

Response 18: Damaged proteins and lipids accumulated in stressed cells alter the structural and potential function of protein adducts, leading to changes in cellular metabolism and functioning in a state of mycotoxicosis.

Point 19: line 435 change “iron ions” into “ferrous ions”, as the meaning is Fe2+ (II)

Response 19: Done

Point 20: line 452-453 Please support your claim “GSH (...) expresses enzymes with antioxidant activity to control.”) and provide a proper reference to the source.

Response 20:

Cells have various mechanisms to prevent ROS-induced damage, which include enzyme systems (SOD, CAT), glutathione liver enzymes, and endogenous antioxidant molecules such as glutathione (GSH). Glutathione is the most important antioxidant synthesized in cells and helps to remove peroxides and many xenobiotic compounds [45]. In addition, GSH is a cofactor and substrate for glutaredoxins (Grx), which catalyze disulfide reductions in the presence of NADPH and glutathione reductase GR. Protein and lipid damage is prevented by enzymes such as GR, multiple glutaredoxins (GSPx), and glutathione-S-transferases (GST). For example, the antioxidant enzyme SOD degrades the superoxide anion radical (•O2ˉ) to H2O and O2, while GPx reduces H2O2 to water and GSH, and the glutathione reductase GR regenerates GSH [46 new].

Point 21: line 448-451 I can’t agree with the description of the function of GST presented here. The reference describes an anti-oxidative action of plant enzymes, whereas mammalian liver GST would be much more fitting. Moreover, the article cited does not use at all words “GST” or “glutathione-S-transferase”.

Response 21:

The antioxidant protection mechanism against peroxides involves glutathione, which is involved in the physiological control of free radical generation and increased oxidative stress through the interaction and expression of GR, GPx, and GST [46, 47 new]. Glutathione reductase (GR) is a homodimeric enzyme that catalyzes the oxidized glutathione (GSSG) reduction process to GSH, using FAD and NADPH as cofactors, thus participating in the maintenance of redox balance in living organisms and helping to remove peroxides and many xenobiotic compounds. Glutathione peroxidases (GPXs) are differentially expressed in cells and tissues and involved in H2O2 detoxification as members of this family reduce peroxides produced by peroxisomes and mitochondria in cells in the cytosol. The activity of glutathione-S-transferases (GST) depends on the presence of GSH, whose role is the reactive electrophilic compounds detoxification, including toxins and products of oxidative stress. The mechanism involves the conjugation of free radicals and non-radical molecules with reduced glutathione GSH conjugated products, which are removed from the cell by specific transporters [48].

Point 22: line 477. logical error in the sentence “The A. indica oil used in the study is not only a phytoremediation plant (...).”

Response 22:

The A. indica oil possesses a phytoremediation capability and acts as a protector that directly inhibits high ROS and RNS, regulates protein oxidation, and collagen deposition, and reduces mycotoxicity OTA.

Point 23: line 486. use “damaged” instead of “induced”

Response 23: done

Bhanwra [56] reported protection by A. indica leaf extract in paracetamol-damaged rats,

Point 24: line 511. The sentence starting with “After protection with ...” should be edited, maybe re-made into two or three shorter sentences. It’s hard to understand your conclusions.

Response 24:

In the group protected with A. indica oil, no degenerative changes in the liver were observed (Figure 1). The cell depletion in the immunocompetent organs of experimental animals treated with a combination of A. indica oil and the OTA showed oil immunostimulatory, hepatoprotective, and immunoprotective properties. Baligar et al. reported a hepatoprotective role of A. indica and its active constituents in hepatotoxicity in rats.

Conclusions In the present study, we reported significantly elevated cytocines, interferon-gamma (IFN-γ) and tumour necrosis factor-alpha (TNF-α), and collagen synthesis, in the OTA-treated mice. Protection with A. indica oil (170 mg/mL) induced cell proliferation of normal hepatocytes and a probable anti-inflammatory therapeutic effect. The results presented above suggest inhibition of OTA-induced inflammation by A. indica oil and ROS reduction influx by regulating IL-1β, IL-6, IL-10, IFN-γ, and TNF-α balance after a 28-day course. OTA is a mycotoxin with a hepatotoxic effect, and toxicity resulted in statistically significant increases in biomarkers of oxidative stress, abnormal hepatic and antioxidant enzymes, and cytokine levels. A. indica oil has a potential restorative and protective effect, as a result of which hepatic and antioxidant enzymes are restored. In addition, the protective effects of A. indica seed oil have led to functional recovery in the OTA-induced mice model. Therefore, A. indica oil can be used as a practical approach to safe OTA-contaminated feed use.

Point 25: Conclusions: 1) Improve language of the manuscript. 2) Present photographs of tissues from animals treated only with neem oil.

Response 25: done

Reviewer 2 Report

In the present manuscript the authors investigate the potential of Azadirachta indica seed oil in reducing Ochratoxine-A hepatotoxicity in a murine model. The introduction and methodology are widely described in this article, although in the presentation of results there are some things that need revision:

1. Figure 1 must have scale bars in the images.

2. Figures 4, 5 and 8 are blurry, quality needs to be improved.

3. In Figures 3-10, the word control and the days appear within the bar of the graph, it is confusing when analyzing the results and sometimes the words do not fit within the total of the bar and are not read correctly. Must be modified.

4. The text in general should be revised to use italics for A. indica (line 68, 115 and 140 among others) and the word in vivo (line 178)

Author Response

Dear reviewer,

Thank you very much for helping us to do our manuscript better.

In the present manuscript, the authors investigate the potential of Azadirachta indica seed oil in reducing Ochratoxin-A hepatotoxicity in a murine model. The introduction and methodology are widely described in this article, although in the presentation of results some things need revision:

Point 1: 1. Figure 1 must-have scale bars in the images.

Response 1: done

Point 2: 2. Figures 4, 5, and 8 are blurry, quality needs to be improved.

Response 2: Done

Point 3: 3. In Figures 3-10, the word control and the days appear within the bar of the graph, it is confusing when analyzing the results and sometimes the words do not fit within the total of the bar and are not read correctly. Must be modified.

Response 3: Done

Point 4: 4. The text, in general, should be revised to use italics for A. indica (lines 68, 115, and 140 among others) and the word in vivo (line 178)

Response 4: Done
